# Recent Therapeutic Advances in Gynecologic Oncology: A Review

**DOI:** 10.3390/cancers16040770

**Published:** 2024-02-13

**Authors:** Elise M. Wilson, Ramez N. Eskander, Pratibha S. Binder

**Affiliations:** Moores Cancer Center, Division of Gynecologic Oncology, Department of Obstetrics, Gynecology, and Reproductive Sciences, University of California San Diego, San Diego, CA 92037, USA; e7wilson@health.ucsd.edu (E.M.W.); reskander@health.ucsd.edu (R.N.E.)

**Keywords:** gynecologic oncology, immunotherapy, antibody–drug conjugates, cervical cancer, endometrial cancer, ovarian cancer

## Abstract

**Simple Summary:**

Novel therapeutic agents have been identified in the management of cervical, endometrial, and ovarian cancer. Immune checkpoint inhibitors improve anti-tumor immunity and have shown clinical efficacy in the management of advanced or metastatic cervical and endometrial cancers. Antibody–drug conjugates couple a potent cytotoxic agent to a highly specific monoclonal antibody to achieve cancer-specific targeted therapy and have proven efficacious in advanced or recurrent cervical, endometrial, and ovarian cancers. The aim of this article is to review recent advances in the targeted treatment of gynecologic malignancies and to review the trials and data underlying the Food and Drug Administration (FDA) approval and National Comprehensive Cancer Network (NCCN) recommendation of these therapeutic agents.

**Abstract:**

Gynecologic malignancies have high incidence rates both nationally and internationally, and cervical, endometrial, and ovarian cancers account for high mortality rates worldwide. Significant research is ongoing to develop targeted therapies to address unmet needs in the field and improve patient outcomes. As tumors mutate and progress through traditional lines of treatment, new therapies must be developed to overcome resistance and target cancer-specific receptors and mutations. Recent advances in the development of immunotherapy and antibody–drug conjugates have resulted in compelling and clinically meaningful results in cervical, endometrial, and ovarian cancers. In the last decade, several immunotherapy agents have received FDA approval or NCCN guideline recommendation for the treatment of gynecologic malignancies, including dostarlimab for advanced or recurrent endometrial cancer and pembrolizumab for advanced or recurrent cervical and endometrial cancers. Several other immunotherapeutic agents are under active investigation. Development of antibody–drug conjugates including tisotumab vedotin in cervical cancer, mirvetuximab soravtansine in ovarian cancer, and trastuzumab deruxtecan in multiple gynecologic cancers has translated into exciting efficacy signals, prompting full drug approvals and additional investigation. This article aims to review recent novel advances in targeted treatments for gynecologic malignancies, highlighting the trials and data underlying these novel interventions.

## 1. Introduction

Gynecologic malignancies account for approximately 11% of newly diagnosed cancers in the United States, and both nationally and globally, cervical, endometrial, and ovarian cancers rank among the most common cancers [1,2]. Cervical and ovarian cancers, in particular, are notable causes of death worldwide [1,2,3]. While most endometrial cancers are diagnosed at early stages, mortality rates have been increasing since 1997 [4]. Treatment for gynecologic malignancies remains a scientific priority, and the development of effective novel treatments in the persistent, advanced, and recurrent settings is needed [5]. Historically, carboplatin (a DNA alkylating agent) and paclitaxel (a microtubule targeting agent) have been the backbone of cytotoxic treatment across gynecologic malignancies [6]. However, most cancers develop mechanisms of resistance to cytotoxic chemotherapy, and effective therapeutic strategies are required. Most novel agents are initially studied in the recurrent disease setting, with transition to earlier lines of treatment if they show sufficient efficacy and improvement in oncologic outcomes with acceptable adverse effect profiles. With the advent of novel targeted technologies, new therapeutic interventions are becoming available that capitalize on the immune system and inherent molecular and genomic signatures of the various gynecologic malignancies. The aim of this paper is to review recent therapeutic advances in the field of gynecologic oncology by focusing primarily on advancements in immunotherapy and antibody–drug conjugates.

## 2. Immune Checkpoint Inhibitors

Programmed cell death protein 1 (PD-1) and its ligands programmed cell death ligand protein 1 (PD-L1) and protein 2 (PD-L2) are checkpoint proteins that regulate the immune response via a regulatory feedback mechanism that limits T-cell expansion and function [7]. These proteins have a key role in tolerance to self-antigens and can calibrate the T-cell response to circulating antigens and tumor cells [8]. PD-L1 is known to be expressed on many different neoplastic cells. The binding of tumor cells expressing PD-L1 to PD-1 on circulating T-cells is an escape mechanism for tumor cells, a way to evade immune recognition and destruction [7]. Overexpression of PD-L1 has been noted in many different tumor cell types [9,10] and is a mechanism of malignant cell survival. Blockade of PD-1 can prevent the interaction of innate T-cell PD-1 to tumor cell-expressed PD-L1, thereby re-establishing T-cell anti-tumor immunity and immune system targeting and thus improving immune-targeted destruction of malignant cells [11]. Research has also evidenced that malignant PD-L1 expression is related to the quality and durability of response to antibody-based PD-1 blockade treatment [12]. Multiple immune checkpoint inhibitors have been developed and are under investigation; several anti-PD-1 agents have been approved by the FDA for the treatment of different malignancies. These immunotherapeutic agents are engineered from one of the four immunoglobulin G isotypes, commonly isotype 1 (IgG1) or isotype 4 (IgG4). IgG1 isotypes can potently activate complement-dependent cytotoxicity (CDC) as well as antibody-dependent cell cytotoxicity (ADCC) and generate strong immunogenic responses. IgG4 isotypes do not activate either pathway and tend to be preferable for indications where potent activation is not desired due to immunogenic reasons [13,14]. Generally, IgG1 isotypes are the most commonly used anti-cancer immunotherapeutics [15]. Here, we will focus on the treatment of gynecologic malignancies.

## 3. Antibody–Drug Conjugates

Antibody–drug conjugates (ADCs) are an emerging class of therapeutic agents that target a cytotoxic “payload” to an intended tumor cell antigen via a stable linker attached to a highly specific monoclonal antibody that binds to a tumor cell of interest [16]. ADCs enable precision targeting of cytotoxic therapies with ideally fewer off-target adverse effects and enhance drug delivery directly to the malignant cells of interest. After the antibody binds to the cell of interest, the ADC is internalized, and the cytotoxic agent is released into the cells, where it can exert its cytotoxic effect [17]. ADCs have become particularly successful in the treatment of breast cancer [18,19,20], and more recently ADCs have been gaining regulatory approvals within gynecologic oncology. Currently, three ADCs are NCCN compendium-listed or FDA-approved for use in gynecologic malignancies, with multiple other agents under investigation with promising results [17].

## 4. Cervical Cancer

### 4.1. Immunotherapeutic Advances in Cervical Cancer

#### 4.1.1. Pembrolizumab

Pembrolizumab is a humanized monoclonal IgG4 kappa anti-PD-1 antibody that has been investigated and approved in multiple malignancies [21,22]. Pembrolizumab targets PD-1, thereby improving anti-tumor immune reactivity. Given its broad mechanism of action, pembrolizumab has proven efficacious in a wide variety of solid tumors and has approval for the management of cervical cancer and endometrial cancer within gynecologic oncology.

Pembrolizumab is given as a 200 mg intravenous infusion every 3 weeks or as a 400 mg intravenous infusion every 6 weeks. The most commonly reported side effects include nausea, fatigue, decreased appetite, constipation, diarrhea, vomiting, stomatitis, and weight loss [23]. Immune checkpoint inhibitors are also associated with immune-related adverse events (irAE), which are a unique spectrum of inflammatory events related to overactivation of the immune system. Immune-related adverse events (irAEs) include, but are not limited to, dermatologic, gastrointestinal, hepatic, endocrine, and inflammatory events.

#### 4.1.2. Biomarkers in Cervical Cancer

KEYNOTE-158 was a phase 2 basket study that evaluated the anti-tumor activity and safety of pembrolizumab in multiple cancer types that were otherwise incurable by surgery, radiation, or chemotherapy [24]. Within this study, patients were also evaluated for the presence of biomarkers that could predict pembrolizumab efficacy. The combined positive score (CPS) was used as a measure of tumor PD-L1 expression and was analyzed by the PD-L1 immunohistochemistry (IHC) 22C3 pharmDx companion diagnostic assay. In the cervical cancer cohort of KEYNOTE-158, 84% of tumors were PD-L1-positive, defined by a CPS ≥ 1, and all clinical responses to pembrolizumab were seen in PD-L1-positive tumors. Consequently, the eligibility for use of pembrolizumab in cervical cancer is based on IHC staining of pathologic specimens against PD-L1, and the CPS must be ≥1 to meet the criteria for treatment with pembrolizumab as monotherapy.

#### 4.1.3. Pembrolizumab Approval in Cervical Cancer

The phase 2 basket study KEYNOTE-158 led to the approval of pembrolizumab monotherapy for the treatment of recurrent or metastatic incurable cervical cancer with CPS ≥ 1 [24]. A subanalysis of the KEYNOTE-158 trial looked specifically at the cervical cancer cohort of 98 patients [25]. In total, 83.7% of the study population was PD-L1-positive with CPS ≥ 1. The primary endpoint, objective response rate (ORR), was 14.6% (95% confidence interval (CI) 7.8% to 24.2%) in the patients with a CPS ≥ 1. This response rate was modestly better than that seen with cytotoxic chemotherapy in the recurrent setting. Importantly, however, for those patients who responded, the median duration of response was not reached, reflecting a provocative efficacy signal in a pretreated cervical cancer population. Subsequently, pembrolizumab monotherapy was approved for use in patients with recurrent PD-L1-positive cervical cancer that had progressed on standard chemotherapy.

Standard first-line cytotoxic treatment for advanced or recurrent cervical cancer was previously a platinum-based chemotherapy and paclitaxel with bevacizumab (an anti-angiogenic agent) based on GOG protocol 240 [26,27]. Pembrolizumab is now incorporated into the frontline treatment of cervical cancer based on the findings from the KEYNOTE-826 trial [28]. This double-blind, placebo-controlled, phase 3 randomized trial enrolled 617 patients, including 548 patients with a PD-L1 CPS ≥ 1. All patients were treated with carboplatin, paclitaxel, with or without bevacizumab (investigator choice), and were randomized to the addition of concurrent treatment with pembrolizumab with pembrolizumab maintenance versus placebo with placebo maintenance. For those with CPS ≥ 1, median progression-free survival (PFS) was 10.4 months in the pembrolizumab arm compared to 8.2 months in the placebo arm (hazard ratio (HR) 0.62; 95% CI 0.50 to 0.77; *p* < 0.001). Twenty-four-month overall survival (OS) was 53% in the pembrolizumab arm and 41.7% in the placebo arm (HR 0.64; CI 0.50 to 0.81; *p* < 0.001). Given the significantly improved PFS and OS endpoints, carboplatin, paclitaxel, and pembrolizumab, with or without bevacizumab, was approved by the US FDA in October 2021 for the first-line treatment of patients with persistent, recurrent, or metastatic cervical cancer.

Pembrolizumab was also examined in patients with newly diagnosed, high-risk, locally advanced cervical cancer—FIGO 2014 stages IB2-IIB with node-positive disease or stages III-IVA. In the KEYNOTE-A18 phase 3 trial, 1060 patients were randomized to pembrolizumab or placebo plus standard chemoradiotherapy (CRT) with cisplatin, external beam radiotherapy, and brachytherapy. An interim analysis of this trial was presented at the European Society for Medical Oncology (ESMO) Congress 2023 [29]. In the pembrolizumab and CRT arm, twenty-four-month PFS was 67.8% compared to 57.3% in the CRT alone arm (HR 0.7; 95% CI 0.55 to 0.89; *p* = 0.002). Overall survival data are not yet mature, but in the interim analysis, the OS favored pembrolizumab over placebo; primary study completion is estimated to be in February 2024. The efficacy signal presented at ESMO led to the US FDA approval of pembrolizumab in high-risk, locally advanced cervical cancer in January 2024. The immunotherapy agent described above is detailed in Table 1.

### 4.2. Antibody–Drug Conjugate Advances in Cervical Cancer

#### 4.2.1. Tisotumab Vedotin

Tisotumab vedotin (TV) is the first ADC approved in a gynecologic malignancy, receiving FDA accelerated approval in September 2021 based on results from the innovaTV 204 study [45]. TV is an ADC comprised of an anti-tissue factor (TF) monoclonal antibody (mAb) bonded to a microtubule-disrupting agent payload, monomethyl auristatin E (MMAE), via a protease-cleavable linker. TF is the main initiator of the extrinsic coagulation pathway and contributes to angiogenesis, progression, and metastasis in many solid tumors [46,47]. High TF expression can be seen in cervical cancer [48]. The anti-TF mAb delivers MMAE to upregulated TF-expressing cervical cancer cells, which then causes direct cytotoxicity and bystander killing of neighboring cells, improving efficacy in heterogenous tumors [47,49]. TV is administered as an intravenous infusion of 2 mg/kg (up to a maximum of 200 mg) every 3 weeks. The most common adverse effects are alopecia, epistaxis, nausea, conjunctivitis, fatigue, and dry eye. The ocular adverse events with TV, resulting in an FDA boxed warning, are due to changes in the corneal epithelium and conjunctiva. Given these ocular side effects, close monitoring with ophthalmology as well as the use of eye care and corticosteroid eye drops during treatment are essential [50].

Prior to TV’s approval, patients with progressive, metastatic cervical cancer had limited options for treatment with poor response rates, in the range of 0–6% for second-line treatments [51,52]. In the innovaTV 204 phase 2 trial [45], 102 patients were enrolled to receive 2 mg/kg of TV intravenously every 3 weeks. The confirmed ORR was the primary endpoint; ORR was 24% in the cohort with 7% complete responses and 17% partial responses. As a result of the durable clinical improvement noted in innovaTV 204, TV received accelerated approval for second-line treatment of recurrent or metastatic cervical cancer, given the poor prognosis for this patient population and the limited efficacy of available chemotherapeutic agents.

The results of the confirmatory innovaTV 301 trial were recently presented at the ESMO 2023 Congress [53]. In this randomized phase 3 trial of TV monotherapy versus investigators’ choice of chemotherapy—topotecan, vinorelbine, gemcitabine, iriniotecan, or pemetrexed—in recurrent or metastatic cervical cancer, 502 patients were randomized. Patients in the TV arm had a 30% reduction in the risk of death compared to the chemotherapy arm, with an HR of 0.70, 95% CI 0.54 to 0.89, and *p* = 0.0038. The median PFS was longer in the TV arm, with an HR of 0.67, 95% CI 0.54 to 0.82, and *p* < 0.0001. The median OS was also significantly longer at 11.5 months for patients treated with TV compared to 9.5 months in the chemotherapy alone arm. This ADC is further detailed in Table 2.

#### 4.2.2. Trastuzumab Deruxtecan

Trastuzumab deruxtecan (T-DXd) is an ADC that uses trastuzumab, a humanized monoclonal antibody against HER2/neu, to deliver a cytotoxic agent to HER2 expressing tumor cells. The cytotoxic agent in T-DXd is deruxtecan, a DNA topoisomerase I inhibitor. Deruxtecan is a potent anti-cancer agent that causes targeted DNA damage and apoptosis. Deruxtecan is also known to have a potent effect on neighboring cells, regardless of HER2 expression, in a phenomenon known as the bystander-killing effect [61]. This enables improved drug efficacy even in tumors with heterogenous HER2 expression.

T-DXd is administered as an intravenous infusion with a dose of 5.4 mg/kg every 3 weeks. Common adverse effects include gastrointestinal disorders, fatigue, anemia, alopecia, decreased appetite, and pain. T-DXd first received accelerated FDA approval in HER2+ breast cancer in 2019, based on the results of DESTINY-Breast01 and 02 [18,67]. The use of T-DXd has since been expanded to tumor site agnostic advanced or metastatic HER2+ expressing cancers in the phase 2 DESTINY-PanTumor02 trial [59]. An interim analysis of the 267 patients enrolled on DESTINY-PanTumor02 was presented at the American Society of Clinical Oncology (ASCO) 2023 Annual Meeting. There was a promising response in the 40-patient cervical cancer cohort, with a 50% ORR [60]. As a result of these data, T-DXd is now listed in the NCCN compendium guidelines for recurrent HER2-positive (IHC 2+ or 3+) cervical cancer [68]. DESTINY-PanTumor02 study completion is estimated to be in March 2027.

## 5. Endometrial Cancer

### 5.1. Immunotherapy Advances in Endometrial Cancer

#### 5.1.1. Biomarkers in Endometrial Cancer

The previously discussed KEYNOTE-158 had an endometrial cancer cohort and a microsatellite instability-high (MSI-H)/mismatch repair deficient (dMMR) solid tumor (except colorectal) cohort. In these two cohorts, 90 patients had MSI-H/dMMR endometrial cancer, and 79 of the 90 had received at least one dose of pembrolizumab with adequate follow-up time for efficacy analysis [69]. Objective response to pembrolizumab was seen in 48% of patients with a median PFS of 13.1 months (95% CI 4.3 to 34.4 months), and median OS and duration of response were not reached. Results of KEYNOTE-158 led to the approval of pembrolizumab monotherapy for dMMR (loss of MMR proteins on IHC staining for MSH-1, MLH-2, MLH-6, and PMS-2) or MSI-H solid tumors as well as in tumor mutational burden-high (TMB-H) solid malignancies [24], regardless of disease site. In the context of the dramatic responses in patients with heavily pretreated recurrent disease, pembrolizumab monotherapy received the first-ever disease site agnostic drug approval by the US FDA for use in recurrent advanced solid tumors based on dMMR or MSI-H tumor status.

#### 5.1.2. Pembrolizumab Approval in Endometrial Cancer

The mainstay of adjuvant treatment for advanced-stage endometrial cancer has been combination chemotherapy with carboplatin and paclitaxel or chemoradiotherapy with the addition of external beam radiation therapy and vaginal brachytherapy to the above regimen, as per GOG protocol 258 [70]. With the approval of pembrolizumab monotherapy in certain endometrial cancer patients, this immunotherapy agent was studied in the frontline setting in combination with chemotherapy and was subsequently added to NCCN compendium-recommended treatment guidelines [31,71]. The NRG GY018 trial was a phase 3 trial of 816 patients with advanced endometrial cancer who were randomized to receive pembrolizumab versus placebo every three weeks in addition to combination chemotherapy with carboplatin and paclitaxel, followed by maintenance pembrolizumab versus placebo [31]. Patients were stratified by dMMR or MMR proficient (pMMR) cohorts. In the dMMR group, median PFS at 12 months in the pembrolizumab arm was 74% compared to 38% in the placebo arm, with an HR of 0.30, 95% CI 0.19 to 0.48, and *p* < 0.001. In the pMMR group, median PFS was 13.1 months in the pembrolizumab arm compared to 8.7 months in the placebo arm, with an HR of 0.54, 95% CI 0.41 to 0.71, and *p* < 0.001. After presentation at the Society of Gynecologic Oncology (SGO) Annual Meeting in 2023 and simultaneous publication, the combination regimen of carboplatin, paclitaxel and pembrolizumab became an NCCN compendium category I-listed recommendation for the primary or adjuvant treatment of stage III-IV endometrial cancer.

Pembrolizumab in combination with lenvatinib, a receptor tyrosine kinase inhibitor, was approved for the treatment of previously treated, recurrent pMMR advanced endometrial cancer as a result of the KEYNOTE-775 trial [30]. In this phase 3 randomized trial of 827 patients, lenvatinib plus pembrolizumab was compared to physician’s choice chemotherapy in patients with advanced, recurrent, or metastatic endometrial cancer, who had progressed after at least one prior platinum-containing chemotherapy regimen. Patients were randomized to once-daily oral lenvatinib with intravenous pembrolizumab every 3 weeks compared to physician’s choice of doxorubicin or single-agent paclitaxel. Median PFS and median OS were significantly longer in the lenvatinib plus pembrolizumab arm compared to the chemotherapy arm. In the pMMR cohort, median PFS was 6.6 months compared to 3.8 months, with an HR of 0.60, 95% CI 0.50 to 0.72, and *p* < 0.001. Median OS was 17.4 months versus 12.0 months, with an HR of 0.68, 95% CI 0.56 to 0.64, and *p* < 0.001. In the overall cohort, including both pMMR and dMMR disease, median PFS was 7.2 months compared to 3.8 months, with an HR of 0.56, 95% CI 0.47 to 0.66, and *p* < 0.001 and median OS was 18.3 months versus 11.4 months, with an HR of 0.62, 95% CI 0.51 to 0.75, and *p* < 0.001. Given that pembrolizumab monotherapy did not have robust responses in pMMR endometrial cancer tumors but the combination of pembrolizumab plus lenvatinib had a significantly longer median PFS and OS, this regimen was approved by the FDA in August 2021 in the advanced and recurrent setting for pMMR endometrial cancer patients.

#### 5.1.3. Dostarlimab Approval in Endometrial Cancer

Dostarlimab-gxly is a humanized monoclonal anti-PD-1 IgG4 antibody that has been studied in multiple dMMR solid tumors [72,73]. Similar to pembrolizumab, dostarlimab works by targeting PD-1 and inhibiting the PD-1 and PD-L1 interaction to prevent tumor immune suppression and thereby improve anti-tumor immune response. Dostarlimab is administered as a 500 mg intravenous infusion every 3 weeks for four–six doses, followed by 1000 mg every 6 weeks for maintenance [74]. The most commonly reported side effects include rash, diarrhea, hypothyroidism, and hypertension [74]. Similar to pembrolizumab, irAEs can be seen with the use of dostarlimab as well.

In 2021, dostarlimab monotherapy received accelerated FDA approval for patients with dMMR recurrent endometrial cancer that has progressed on or followed platinum-based chemotherapy. A phase 1 non-randomized trial evaluating the clinical activity and safety of dostarlimab in patients with dMMR recurrent endometrial cancer showed an ORR of 42.3% (95% CI 30.6% to 54.6%) with a 12.7% complete response rate and 29.6% partial response rate [75]. Importantly, responses were durable, and the median duration of response was not reached after a median follow-up time of 11.2 months.

In 2023, dostarlimab in combination with standard carboplatin and paclitaxel was approved for use in advanced dMMR endometrial cancer based on the results of the RUBY trial [33]. In this phase 3 trial, 494 patients with primary advanced or first recurrent endometrial cancer were randomized to 500 mg dostarlimab or placebo plus carboplatin and paclitaxel every 3 weeks, followed by 1000 mg dostarlimab or placebo every 6 weeks for up to 3 years for maintenance. Stratification was performed based on MMR status, and patients with dMMR/MSI-H disease were studied as a specific cohort. In the dMMR/MSI-H cohort, twenty-four-month PFS was 61.4% in the dostarlimab arm versus 15.7% in the placebo arm, with an HR of 0.28, 95% CI 29.3 to 42.9, and *p* < 0.001. In the overall intent-to-treat population, twenty-four-month PFS was 36.1% in the dostarlimab arm compared to 18.1% in the placebo arm, with an HR of 0.64, 95% CI 0.51 to 0.80, and *p* < 0.001. Twenty-four-month OS was 71.3% in the dostarlimab arm compared to 56.0% in the placebo arm, with an HR of 0.64, 95% CI 0.46 to 0.87, *p* = 0.0021. Given the significant PFS benefit in the dMMR endometrial cancer cohort, dostarlimab plus chemotherapy was approved by the FDA in July 2023 for use in patients with dMMR advanced or recurrent endometrial cancer in the frontline settling.

#### 5.1.4. Durvalumab

Durvalumab is a human IgG1 kappa monoclonal antibody against PD-L1. Similar to pembrolizumab and dostarlimab, durvalumab works by improving the anti-tumor immune response by blocking the PD-1 and PD-L1 interaction [76]. Durvalumab is currently approved in several solid tumors [77,78] and has been examined in endometrial cancer. Durvalumab is administered as a 1120 mg infusion every 3 weeks for six cycles, followed by 1500 mg every 4 weeks for maintenance therapy. The most commonly reported side effects are cough, fatigue, pneumonitis, dyspnea, and rash [79]. Immune-related AEs can be seen with the use of durvalumab as well.

The DUO-E randomized phase 3 trial evaluated the clinical activity of the addition of durvalumab to concurrent chemotherapy and as maintenance, as well as the combination of durvalumab and olaparib maintenance [34]. Olaparib is a poly(ADP-ribose) polymerase inhibitor (PARPi) that is hypothesized to enhance the response of checkpoint inhibitor immunotherapies via synthetic lethality [80]. In synthetic lethality, immune checkpoint inhibitors and PARPi work synergistically to increase DNA damage, disrupt DNA repair, and, therefore, increase cancer cell cytotoxicity [81,82]. A total of 713 patients were randomized to three arms stratified by MMR status (dMMR versus pMMR), disease status (newly diagnosed versus recurrent), and geographic location (Asia versus non-Asia). All patients received standard carboplatin and paclitaxel; one arm received the addition of concurrent placebo plus placebo maintenance, one arm received concurrent durvalumab plus durvalumab maintenance and placebo maintenance, and one arm received concurrent durvalumab plus durvalumab and olaparib maintenance. In the intention-to-treat population, median PFS was improved in the durvalumab arm (HR 0.71; 95% CI 0.57 to 0.89; *p* = 0.003) and in the durvalumab plus olaparib arm (HR 0.55; 95% CI 0.43 to 0.69; *p* < 0.0001) compared to the control arm. PFS benefit was seen in the dMMR and pMMR subgroups as well as the PD-L1-positive subgroup, with interim overall survival results supportive of the durvalumab arms. In the durvalumab arm, the interim (approximately 28% mature) median OS HR was 0.77 (95% CI 0.56 to 1.07; *p* = 0.12), and in the durvalumab plus olaparib arm, the interim median OS HR was 0.59 (95% CI 0.42 to 0.83; *p* = 0.003). Durvalumab has not yet been NCCN compendium listed or FDA approved for uterine cancer [71], and the DUO-E trial is ongoing, with primary trial completion estimated to be in March 2025.

#### 5.1.5. Atezolizumab

Similar to durvalumab, atezolizumab is a humanized anti-PD-L1 IgG1 monoclonal antibody that improves anticancer immunity and is approved in several solid malignancies [83,84]. Atezolizumab is administered as a 1200 mg infusion every 3 weeks for six to eight cycles, followed by the same dosing for maintenance therapy. The most common adverse events include fatigue, anorexia, nausea, cough, and dyspnea [85]. Immune-related AEs can be seen with the use of atezolizumab as well.

Atezolizumab is currently under investigation in the AtTEnd trial, with interim results having been presented at the ESMO Congress 2023 [37]. AtTEnd is a phase 3 randomized trial in patients with advanced or first recurrent endometrial cancer or carcinosarcoma. dMMR and pMMR patients were included. Patients were randomized into two arms: carboplatin, paclitaxel, and atezolizumab plus atezolizumab maintenance versus carboplatin, paclitaxel, and placebo plus placebo maintenance. In total, 22.8% of the population had dMMR tumors. In the dMMR population, median PFS was improved in the atezolizumab arm compared to the placebo arm (HR 0.36; 95% CI 0.23 to 0.57; *p* = 0.0219). In the entire cohort, improved median PFS was again seen, with an HR of 0.74, 95% CI 0.61 to 0.91, and *p* = 0.0219. The AtTEnd trial is ongoing for OS and outcome data, with study completion estimated to be in May 2025, and atezolizumab is not yet NCCN compendium-listed or FDA-approved for endometrial cancer [71].

### 5.2. Monoclonal Antibodies and Antibody–Drug Conjugate Advances in Endometrial Cancer

#### 5.2.1. Trastuzumab

Trastuzumab, a humanized monoclonal antibody against HER2/neu, was first approved for HER2+ breast cancer as a targeted therapy in 1998 [86]. HER2 is a receptor tyrosine kinase known to be overexpressed in several solid tumors, including endometrial and ovarian cancers [87,88]. Trastuzumab binds to HER2 and prevents HER2-mediated signalling. Serous endometrial cancer is known to have significant upregulation of HER2+, making trastuzumab an attractive target in this subset of patients [89]. Serous endometrial cancer is an aggressive type of endometrial cancer with high mortality rates and poor prognosis, hence the compelling need for novel treatment options in this cancer type [90]. A phase 2 single-arm trial evaluating the addition of trastuzumab to conventional carboplatin and paclitaxel compared to carboplatin and paclitaxel alone was performed, and results were published in 2018 [60]. The study showed an improved median PFS of 17.9 months in the trastuzumab arm compared to 9.3 months in the chemotherapy-alone arm (HR 0.40; 90% CI 0.20 to 0.80; *p* = 0.013) [91]. Trastuzumab is administered with an 8 mg/kg loading dose, followed by 6 mg/kg every 3 weeks, in addition to standard administration of carboplatin and paclitaxel followed by maintenance until toxicity or disease progression. The most significant side effects included hypertension, anemia, and neutropenia. The compelling results of the trial resulted in the approval of trastuzumab for treatment of advanced serous endometrial cancer in addition to carboplatin and paclitaxel.

#### 5.2.2. Trastuzumab Deruxtecan

T-DXd is under active investigation in endometrial cancer. The phase 2 DESTINY-PanTumor02 trial includes an endometrial cancer cohort of 40 patients [59]. In the interim trial analysis presented at the ASCO 2023 Annual Meeting, a particularly robust response was seen in the endometrial cancer cohort, with a 57.5% ORR [60]. Given a response rate much higher than would be anticipated with cytotoxic chemotherapy in this pretreated population, T-DXd is now listed in the NCCN compendium guidelines [71].

T-DXd was also studied in HER2+ advanced or recurrent uterine carcinosarcoma in the STATICE trial [61]. Approximately one third of uterine carcinomas are known to be HER2+ [92], and T-DXd is a promising targeted agent in this rare malignancy with a poor prognosis and high mortality rate [93]. In this single-arm phase 2 trial, 32 patients were included in the analysis and were stratified by HER2-high (IHC staining 2+ or 3+) or HER2-low (IHC staining 1+) status. ORR was 54.5% (95% CI 32.2 to 75.6%) in the HER2-high cohort and 70% (95% CI 34.8 to 93.3%) in the HER2-low exploratory analysis, with a 100% disease control rate across the entire population. The median PFS and OS in the HER2-high cohort were 6.2 months and 13.3 months. Interestingly, the median PFS and OS was 6.7 months and was not reached in the HER2-low cohort. The high response rate in the HER2-low cohort may be explained in part by the bystander effect on heterogenous tumors. This trial provided promising results for the efficacy of T-DXd in this rare and aggressive uterine cancer subtype.

## 6. Ovarian Cancer

### 6.1. Immunotherapy in Ovarian Cancer

#### 6.1.1. Biomarkers in Ovarian Cancer

The previously discussed KEYNOTE-158 trial included a 15-patient ovarian cancer cohort [24]. Patients in this cohort had an ORR of 33.3% and a median PFS of 2.3 months. Although not leading to a specific ovarian cancer indication, KEYNOTE-158 led to the approval of pembrolizumab in dMMR, MSI-H, or TMB-H solid malignancies in a disease site agnostic fashion and, therefore, is approved in patients with dMMR, MSI-H, or TMB-H recurrent ovarian cancer. Similarly, the GARNET trial included a small ovarian cancer cohort of seven patients [72]. These patients had an ORR of 42.9%. A median PFS was not calculated given the small cohort size. Overall, the GARNET trial led to FDA approval for dostarlimab in advanced or recurrent dMMR solid tumors, leading to an approval for recurrent ovarian cancer patients with dMMR disease.

#### 6.1.2. Immunotherapy Trials in Ovarian Cancer

The phase 3 IMagyn050 trial compared atezolizumab plus carboplatin, paclitaxel, and bevacizumab versus carboplatin, paclitaxel, and bevacizumab alone in patients with newly diagnosed stage III-IV ovarian cancer [38]. The median PFS was not met in the intention-to-treat population (HR 0.92; 95% CI 0.79 to 1.07; *p* = 0.28) or in the PD-L1-positive population (HR 0.80; 95% CI 0.65 to 0.99; *p* = 0.038), and interim OS survival results showed no benefit from the addition of atezolizumab. In a separate non-randomized phase 2 trial looking at pembrolizumab plus bevacizumab and oral metronomic cyclophasphamide (a DNA alkylating agent) in patients with recurrent ovarian cancer, ORR was 47.5%, with clinical benefit seen in 95% of patients, and the durable treatment response rate was 25% [32]. The phase 2 basket MEDIOLA trial is currently examining the role of olaparib plus durvalamab in several solid tumor cohorts, including in germline BRCA mutated (gBRCAm) platinum sensitive ovarian cancer [35]. A expansion cohort analysis of the MEDIOLA trial, looking at durvalumab and olaparib plus or minus bevacizumab, was recently released, looking at the gBRCAm patients as well as two new cohorts of non-gBRCAm platinum-sensitive relapsed ovarian cancer, plus or minus bevacizumab [36]. The gBRCAm cohort ORR was 92.2% (95% CI 81.1 to 97.7%) and the twenty-four-week disease control rate was 88.2% (90% CI 78.1 to 94.8). ORR in the non-gBRCAm with bevacizumab arm was 87.1% (95% CI 70.2 to 96.4), and the twenty-four-week disease control rate was 74.2% (90% CI 58.2 to 86.5%). The ORR in the gBRCAm without bevacizumab arm was 34.4% (95% CI 18.6 to 53.2), and the twenty-four-week disease control rate was 28.1% (90% CI 15.5 to 43.9). The above-listed trials are not NCCN compendium-listed or FDA-approved. Overall, the only immunotherapy indications in recurrent ovarian cancer currently NCCN compendium-listed and FDA-approved are pembrolizumab or dostarlimab for dMMR, MSI-H or TMB-H biomarker-positive recurrent ovarian cancer.

### 6.2. Antibody–Drug Conjugate Advances in Ovarian Cancer

#### 6.2.1. Mirvetuximab Soravtansine

Mirvetuximab soravtansine (MIRV) was the second ADC that received FDA approval in gynecologic malignancies and was the first approved for ovarian carcinoma. Up to 80% of ovarian cancer patients will recur, and secondary lines of treatment are dependent on time since last exposure to platinum-containing chemotherapy [94]. Many patients will ultimately develop platinum-resistant ovarian cancer (PROC), whereupon they progress within 6 months of receiving a platinum-containing chemotherapy regimen. Therapeutic options are limited at this stage, making PROC a significant area of interest in targeted therapy research [95,96].

MIRV is an ADC comprised of a humanized antifolate receptor α (FRα) mAb attached to an anti-tubulin maytansinoid DM4 payload via a cleavable linker [97]. After FRα mAb binding, linker cleavage, and intracellular transport and accumulation, DM4 acts as a potent anti-mitotic agent [98]. FRα is responsible for the transport of folate into cells and is known to be highly expressed in epithelial ovarian cancers, particularly in high-grade serous ovarian cancer, making FRα an attractive target in PROC [99]. FRα expression can be identified by IHC percent positivity, with ≥25% tumor staining at 2+ intensity indicating FRα positivity [100]. MIRV is given as an intravenous infusion of 6 mg/kg adjusted ideal body weight every 3 weeks. Due to ocular toxicities similar to TV, resulting in an FDA boxed warning, close follow-up with ophthalmology and prophylactic use of ocular corticosteroids and lubricants is indicated [101].

MIRV received accelerated FDA approval in November 2022, based on the results of the SORAYA trial [56]. In SORAYA, a single-arm phase 2 study, 106 PROC patients were enrolled. Eligibility was limited to patients who had received three prior lines of therapy or less. The primary endpoint, ORR, was 32.4% (95% CI 23.6 to 42.2; *p* < 0.001), with five complete and twenty-nine partial responses. The median duration of response was 6.9 months. The confirmatory MIRASOL trial, a randomized phase 3 trial comparing MIRV to standard-of-care chemotherapy, has completed accrual, with finalized results and publication pending. Study completion is estimated to be in April 2024. An interim analysis of the trial was presented at the ASCO 2023 Annual Meeting [57]. The median PFS in the MIRV arm was 5.62 months compared to 3.98 months in the chemotherapy arm, with an HR of 0.64, 95% CI 0.49 to 0.84, and *p* < 0.0001. An OS benefit was also seen, with a median OS of 16.5 months in the MIRV arm compared to 12.7 months in the chemotherapy arm (HR 0.67; 95% CI 0.5 to 0.88; *p* = 0.0046). MIRV is being evaluated in combination with other therapies in several ongoing clinical trials. The GLORIOSA trial is evaluating the role of MIRV plus bevacizumab versus bevacizumab alone as maintenance in patients with recurrent platinum-sensitive ovarian cancer. The trial is currently enrolling, and the trial design was presented at the ASCO 2023 Annual Meeting [58]. Study completion is estimated to be in March 2027. The IMGN853-0420 phase 2 study is evaluating the combination of carboplatin and concurrent MIRV followed by MIRV maintenance in patients with recurrent, FRα-positive platinum sensitive ovarian cancer, and is currently enrolling, with study completion estimated to be in June 2024 [59].

#### 6.2.2. Trastuzumab Deruxtecan

Within the DESTINY-PanTumor02 trial, 40 patients with ovarian cancer were treated with T-DXd monotherapy. In the interim analysis presented at ASCO 2023, patients in the ovarian cancer cohort also had a high ORR of 45.0%, making this a promising agent in ovarian cancer [60]. This investigation is still ongoing and T-DXd is not currently listed in NCCN compendium guidelines for the treatment of ovarian cancer [95].

## 7. Conclusions

Multiple novel therapeutic agents have been developed for gynecologic malignancies in recent years, capitalizing on unique molecular markers innate to the different malignancies. Advancements in immunotherapy and ADCs have broadened treatment options and improved outcomes for patients with advanced, recurrent, or metastatic cervical, endometrial, and ovarian cancers. The approved treatments are overall well tolerated, and treating physicians have become well accustomed to altering treatment doses or treatment breaks depending on toxicities. Pembrolizumab, dostarlimab, durvalumab, and atezolizumab act as immune checkpoint inhibitors and improve the immune system’s ability to recognize and kill malignant cells. ADCs such as TV, MIRV, and TDXd link a potent cytotoxic agent to a highly specific monoclonal antibody for better drug delivery and minimization of systemic toxicities.

The development of new therapeutics continues at a rapid pace. Multiple novel immunotherapy drugs, including the anti-CTLA4 agent ipilimumab, the anti PD-1 agent nivolumab, and the anti-PD-L1 agent avelumab, are being evaluated as monotherapy and in combination with other agents. Similarly, ADCs targeting FRα, HER2, and Trop2 are under investigation [17,102] and are being evaluated as monotherapy or in combination with other agents. Newer technologies are also undergoing investigation, including nanobodies, chimeric antigen receptor T-cells, and single-domain antibody fragments [103,104,105]. Development in these technologies is nascent in gynecologic malignancies. The targeted treatment of gynecologic malignancies will continue to evolve and develop as novel agents with improved response rates, higher specificity, and fewer off-target effects are developed.

## Figures and Tables

**Table 1 cancers-16-00770-t001:** Recent Immunotherapy Advances in gynecologic oncology.

Agent	Target	Malignancy	Clinical Trial	Approval Year
Pembrolizumab	PD-1	Cervix	KEYNOTE-158 [25]	2021
			KEYNOTE-826 [28]	
			KEYNOTE-A18 [29]	
		Uterine	KEYNOTE-775 [30]	2021
			GY018 [31]	
		Ovary	NCT02853318 [32]	N/A
Dostarlimab	PD-1	Uterine	RUBY [33]	2023
Durvalumab	PD-L1	Uterine	DUO-E [34]	N/A
		Ovary	MEDIOLA [35,36]	N/A
Atezolizumab	PD-L1	Uterine	AtTEnd [37]	N/A
		Ovary	IMagyn050 [38]	N/A
Cemiplimab	PD-1	Cervix	GOG-3016 [39]	NCCN compendium listed
Ipilimumab + Nivolumab	CTLA4 + PD-1	Uterine	NCI-2021-11881 [40]	N/A
		Ovary	NCI-2014-02424 [41]	N/A
		Ovary, Uterine, and Cervix	2017-0264 [42]	N/A
Avelumab	PD-L1	Uterine	16-322 [43]	N/A
			MITO END-3 [44]	

**Table 2 cancers-16-00770-t002:** Recent ADC advances in gynecologic oncology.

Agent	Antibody Target	Payload	Payload Mechanism of Action	Malignancy	Clinical Trial	Approval Year
Tisotumab vedotin	TF	Monomethyl auristatin E (MMAE)	Anti-microtubule agent	Cervix	innovaTV 204 [45]	2021
					innovaTV 301 [53]	
Mirvetuximab soravtansine	FRα	DM4	Anti-tubulin maytansinoid agent	Uterine	2000023841 [54]	N/A
					18-602 [55]	
				Ovary	SORAYA [56]	2022
					MIRASOL [57]	
					GLORIOSA [58]	
					IMGN853-0420 [59]	
Trastuzumab deruxtecan	HER2/neu	Deruxtecan	DNA topoisomerase I inhibitor	Cervix	DESTINY-PanTumor02 [60]	NCCN compendium listed
				Uterine	STATICE [61]	NCCN compendium listed
					DESTINY-PanTumor02 [60]	
				Ovary	DESTINY-PanTumor02 [60]	N/A
Sacituzumab govitecan	Trop2	SN-38	DNA topoisomerase I inhibitor	Cervix	2000023639 [62]	N/A
				Uterine	2000026850 [63]	N/A
				Ovary	2000036114 [64]	N/A
Luveltamab tazevibulin	FRα	SC209	Hemiasterlin	Ovary	REFRaME [65]	N/A
Farletuzumab ecteribulin	FRα	Eribulin	Microtubule inhibitor	Ovary	CA116-001 [66]	N/A

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
