# Peer review of "Recent Therapeutic Advances in Gynecologic Oncology: A Review"

_cancers, 2024, doi:10.3390/cancers16040770_

Round 1
Reviewer 1 Report
Comments and Suggestions for Authors
This kind of review is important for women's health, but I have a few questions:
1.Your topic is Recent Therapeutic Advances in Gynecologic Oncology: A Review.When it comes to women's cancers, why breast cancer, which has the most widespread impact on women's health, is not in the scope of the discussion.At the very least, you should include breast cancer in the discussion
2.You've discussed Biomarkers in Immunotherapy in every disease, and that's good.Why is it missing when discussing Ovarian Cancer, please explain.
3.As a review, your conclusion or discussion section is essential, and it seems a little too short right now. And you need to increase the number of references as a review.
4.You're focusing on targeted therapy or immunotherapy, but your introduction to immunotherapy and immune checkpoints is too simplistic, and you should add a description of this to the introduction.
Minor editing of English language required.
Author Response
Dear Reviewer,
We appreciate your review of our manuscript. Our responses to your comments and suggestions are included below.
1. Your topic is Recent Therapeutic Advances in Gynecologic Oncology: A Review. When it comes to women's cancers, why breast cancer, which has the most widespread impact on women's health, is not in the scope of the discussion. At the very least, you should include breast cancer in the discussion.
We appreciate the reviewer’s question. For the purposes of this review, we are focusing specifically on the gynecologic malignancies, and not all women’s cancer. We believe breast cancer is outside the scope of a gynecologic malignancy review article and is not treated by gynecologic oncologists, but we agree it is appropriate to include in other reviews addressing all female cancers.
2. You've discussed Biomarkers in Immunotherapy in every disease, and that's good. Why is it missing when discussing Ovarian Cancer, please explain.
We regret the confusion. We have now specifically included a Biomarkers in Ovarian Cancer section (section 6.1.1), when previously it was included within a general “Immunotherapy in Ovarian Cancer” section, as there is a paucity of indications for immunotherapy in ovarian cancer.
3. As a review, your conclusion or discussion section is essential, and it seems a little too short right now. And you need to increase the number of references as a review.
We regret the confusion. As per Cancers template, the Conclusions section is “mandatory, with one or two paragraphs to end the main text.” We believe our 2 paragraph conclusion appropriately meets the journal expectations, and we have included extended discussion throughout the review. We have also added to the conclusion paragraph and included more detail and references on newer technologies that are early in development.
We note the reviewer’s comment and have added an additional 8 references. We believe over 100 references is appropriate for the scope of this review, and is reflective of the number of references in similar review papers within the field.
4. You're focusing on targeted therapy or immunotherapy, but your introduction to immunotherapy and immune checkpoints is too simplistic, and you should add a description of this to the introduction.
We regret the confusion. In lines 23-46 we have written an introduction to immune checkpoint inhibitors and immunotherapy, and have included additional information on the IgG isotypes. As this paragraph is the next section after the introduction paragraph, we believe this is an appropriate place to provide a longer description of the mechanism of action, leaving the introduction as a brief introduction to the entirety of the paper. We have now moved up the mechanism of action of ADCs (lines 48-65) to immediately follow the mechanism of actions of immunotherapy section. Overall, this is now formatted as an extended introduction with discussion of mechanisms of action of the therapeutic types of interest immediately following the introductory paragraph.
Reviewer 2 Report
Comments and Suggestions for Authors
The document cancers-2846618 is a review that presents the latest advancements in targeted treatments for gynecologic malignancies, emphasizing the trials and data supporting these new interventions. It is scientifically robust, well-organized, and pleasant to read. The manuscript is informative in its current form The reviewer offers suggestions for improvement.
1. For consistency: indicate mAb isotypes for all immunotherapies. Provide a brief comment on the advantages and disadvantages of IgG1 versus IgG4 isotypes.
2. L282: Briefly explain synthetic lethality.
3. For ongoing clinical trials: specify the estimated date of study completion.
4. Adding author personal views would enhance the manuscript. Consider discussing costs and QALY in general.
5. Consider exploring the development of newer technologies, such as perspectives on the use of nanobodies or single domain antibody fragments (e.g., NCT04467515 phase 1 on CAM-Her2). The referee is not related to this RCT.
Comments on the Quality of English LanguageRectify minor typos, such as l376: "an expansion cohort."
Address minor abbreviation problems, like l200: "CRT" and l445-446: abbreviate drug names.
Adjust reference style to adhere to Cancers journal's instructions.
Author Response
Dear Reviewer,
We appreciate your review of our manuscript, and have included our responses to your comments below.
1. For consistency: indicate mAb isotypes for all immunotherapies. Provide a brief comment on the advantages and disadvantages of IgG1 versus IgG4 isotypes.
We regret the confusion. We added the isotype for the immunotherapeutic agent atezolizumab and confirmed mAb isotypes were included for all other immunotherapeutic agents listed in the body of the manuscript. We appreciate the reviewer’s comment and have added a segment on IgG1 vs. IgG4 isotypes into lines 39-45.
2. L282: Briefly explain synthetic lethality.
We appreciate the reviewer’s comment. We have added in a description of synthetic lethality as recommended to this section.
3. For ongoing clinical trials: specify the estimated date of study completion.
We appreciate the reviewer’s comment. We have added in estimated dates of study completion for ongoing clinical trials.
4. Adding author personal views would enhance the manuscript. Consider discussing costs and QALY in general.
We appreciate the reviewer’s comment. For the purposes of this manuscript, we prefer to keep a neutral perspective and report the literature that is currently available as it pertains to response and efficacy of the described drugs, rather than focusing on cost and QALY for the purposes of this review paper.
5. Consider exploring the development of newer technologies, such as perspectives on the use of nanobodies or single domain antibody fragments (e.g., NCT04467515 phase 1 on CAM-Her2). The referee is not related to this RCT.
We appreciate the reviewer’s recommendation. For the purposes of this review we have chosen to focus on immunotherapeutic agents and antibody-drug conjugates that have largely undergone phase 2 or 3 trials within the last several years. We have included a sentence in the conclusion (lines 538-542) referencing that newer technologies are also undergoing development and investigation, but are very early in development in the gynecologic malignancies, and so we will not be focusing on these for the purposes of this review.
6. Rectify minor typos, such as l376: "an expansion cohort."
We note the reviewer’s suggestion. The use of “expansion cohort” was intentional, and echoes the language used in the referenced paper. The referenced trial used an expanded cohort of patients from an earlier trial design that has not yet been published.
7. Address minor abbreviation problems, like l200: "CRT" and l445-446: abbreviate drug names.
We note the reviewer’s suggestion. “CRT” is defined in line 132 as chemoradiotherapy, and is used as the abbreviation CRT in line 135. We note the reviewer’s suggestion for line 445-446 and have abbreviated the drug names accordingly.
8. Adjust reference style to adhere to Cancers journal's instructions.
We note the reviewer’s comment. We have reviewed the references and adjusted accordingly to adhere to the stated journal specifications.